# Health-Related Quality of Life and Mental Health after Surgical Treatment of Hepatocellular Carcinoma in the Era of Minimal-Invasive Surgery: Resection versus Transplantation

**DOI:** 10.3390/healthcare9060694

**Published:** 2021-06-09

**Authors:** Linda Feldbrügge, Alexander Langenscheidt, Felix Krenzien, Mareike Schulz, Nicco Krezdorn, Kaan Kamali, Andreas Hinz, Michael Bartels, Panagiotis Fikatas, Moritz Schmelzle, Johann Pratschke, Christian Benzing

**Affiliations:** 1Department of Surgery, Campus Charité-Mitte and Campus Virchow Klinikum, Charité-Universitätsmedizin Berlin, Corporate Member of Freie Universität Berlin and Humboldt-Universität zu Berlin, 13353 Berlin, Germany; alexander.langenscheidt@charite.de (A.L.); felix.krenzien@charite.de (F.K.); mareike.schulz@charite.de (M.S.); kaan.kamali@charite.de (K.K.); panagiotis.fikatas@charite.de (P.F.); moritz.schmelzle@charite.de (M.S.); johann.pratschke@charite.de (J.P.); christian.benzing@charite.de (C.B.); 2Department of Plastic, Hand and Reconstructive Surgery, Hannover Medical School, 30625 Hannover, Germany; krezdorn.nicco@mh-hannover.de; 3Department of Medical Psychology and Medical Sociology, University Hospital Leipzig, 04103 Leipzig, Germany; andreas.hinz@medizin.uni-leipzig.de; 4Department of General, Visceral and Thoracic Surgery, Helios Park-Klinikum Leipzig, 04103 Leipzig, Germany; michael.bartels@helios-gesundheit.de

**Keywords:** hepatocellular carcinoma, minimally invasive liver surgery, health-related quality of life, liver transplantation

## Abstract

Laparoscopic liver resection (LLR) is an increasingly relevant treatment option for patients with resectable hepatocellular carcinoma (HCC). Orthotopic liver transplantation (OLT) has been considered optimal treatment for HCC in cirrhosis, but is challenged by rising organ scarcity. While health-related quality of life (HRQoL) and mental health are well-documented after OLT, little is known about HRQoL in HCC patients after LLR. We identified all HCC patients who underwent LLR at our hospital between 2014 and 2018. HRQoL and mental health were assessed using the Short Form 36 and the Hospital Anxiety and Depression Scale, respectively. Outcomes were compared to a historic cohort of HCC patients after OLT. Ninety-eight patients received LLR for HCC. Postoperative morbidity was 25% with 17% minor complications. LLR patients showed similar overall HRQoL and mental health to OLT recipients, except for lower General Health (*p* = 0.029) and higher anxiety scores (*p* = 0.010). We conclude that LLR can be safely performed in patients with HCC, with or without liver cirrhosis. The postoperative HRQoL and mental health are comparable to that of OLT recipients in most aspects. LLR should thus always be considered an alternative to OLT, especially in times of organ shortage.

## 1. Introduction

Any chronic injury to the liver, including viral hepatitis, alcoholic and non-alcoholic fatty liver disease, can lead to liver fibrosis with liver cirrhosis as the final stage. The majority of hepatocellular carcinomas (HCC) develop in cirrhotic livers or following viral hepatitis [1]. Besides predisposition for HCC, liver cirrhosis has other serious complications that often greatly influence quality of life [2,3]. HCC treatment with curative intent can consist of surgical resection, local ablation, or orthotopic liver transplantation (OLT) [1]. Orthotopic liver transplantation (OLT) has been argued to be the therapy of choice for patients with HCC and underlying cirrhosis [4], as it cures patients from both the tumor and cirrhosis with its complications. In case of surgically resectable HCC foci and contraindications for OLT, liver resection can be an alternative, but has previously been associated with high postoperative morbidity and mortality in cirrhotic patients [5]. However, liver transplantation and life-long immunosuppression also bring along further risks and side-effects that have been shown to effect health-related quality of life (HRQoL)—we have previously shown that despite very good physical long-term outcomes, recipients of liver transplants display reduced quality of life, for example higher levels of fatigue, when compared to the general population [6,7,8]. Furthermore, psychological distress, including anxiety and depression, significantly affects the mental health of OLT recipients [6,9].

In the course of the modernization of surgery with the introduction of improved perioperative management concepts and, above all, minimally-invasive surgical techniques, morbidity and mortality of liver resection could be drastically reduced [10,11,12]. Furthermore, current studies show a significantly improved overall survival and disease-free survival after minimally invasive liver resection for HCC with underlying liver cirrhosis, establishing it as an equivalent alternative to liver transplantation for many patients [13]. This becomes even more important when considering the increasing shortage of donated organs [14]. In Germany, the number of organ donors decreased by 30% over the course of nine years [15,16].

Due to the ever-improving clinical results after both liver resection and liver transplantation, the assessment of HRQoL has become a central outcome factor [7,17,18,19]. HRQoL after liver resection for HCC is not yet sufficiently understood, especially in the era of minimally invasive surgery. Potential advantages of laparoscopic or robotic liver resection compared to OLT (for example, no need for immunosuppression, minor surgical trauma) are possibly offset by disadvantages such as the persistence of the underlying cirrhosis [13,20,21].

The present study seeks to evaluate the HRQoL and mental health of HCC patients who underwent minimally invasive surgical resection. Furthermore, these results are compared to the HRQoL and mental health of patients who underwent OLT for HCC. 

## 2. Materials and Methods

### 2.1. Study Design and Patients

All patients who underwent laparoscopic liver resection (LLR) for HCC at our hospital between June 2014 and December 2018 were included in the study. The search algorithm via the hospital information system included filtering by diagnosis (HCC) and procedure codes (anatomical and non-anatomical liver resections). Further inclusion criteria were: age > 18 years, and complete patient record. Patients were excluded if they had merely received open liver biopsy or in case of multivisceral resections, and if they lived outside of Germany or did not speak German. In total, 98 HCC patients who underwent minimally-invasive liver resection were included in the study for analysis of clinical outcome parameters.

HRQoL and mental health data were obtained in two surveys that were conducted at two different time points: One was carried out in March and April 2019 and included patients who had received surgery between November 2016 and December 2018. In total, 67 patients who underwent minimally-invasive liver resection for HCC were identified; 30 (45%) returned the questionnaire. Patients who had undergone laparoscopic liver resection between June 2014 and October 2016 had already been interviewed in an earlier HRQoL survey about laparoscopic liver surgery, with results published in parts previously [22]. From this study period, 31 HCC patients were identified, with questionnaires available for 13 of these patients (42%). 

HRQoL and mental health scores of HCC patients after LLR were then compared to patients who underwent OLT for HCC. These patients were part of a large cohort of liver transplant patients whose HRQoL data had been collected and partly published earlier by our group [6,8,23]. In a second step, we compared results from HCC patients after LLR with HRQoL data from a subset of the German general population (German health survey which took place between 1997 and 1999), that was matched with our patient cohort according to age and gender [24].

We assessed HRQoL and mental health using the Short Form 36 (SF-36) and the Hospital Anxiety and Depression Scale, respectively. The surveys including return envelopes free of charge were sent to the patients via mail. All patients who had not answered our survey within two weeks were contacted again. The study was approved by the local ethics committees.

### 2.2. Health-Related Quality Assessment Using the Short Form 36 (SF-36)

The Short Form 36 (SF-36) has frequently been used to assess the HRQoL of patients [25,26]. The SF-36 is a questionnaire containing 36 items about the person’s health status, which is divided into the following eight HRQoL dimensions: Physical Functioning (PF, ten items), Role-Physical (RP, four items), Bodily Pain (BP, two items) and General Health (GH, six items), Vitality (VIT, four items), Social Functioning (SF, two items), Role-Emotional (RE, three items), and Mental Health (MH, five items). Two factors of second order can be calculated: Physical Component Summary (PCS) and Mental Component Summary (MCS). The possible scores in the eight subscales range between 0 and 100 points with 0 points meaning the worst and 100 points meaning the best HRQoL outcome. 

### 2.3. Hospital Anxiety and Depression Scale (HADS)

The HADS is a survey used to detect and measure depression and anxiety. It represents an established mental health assessment instrument developed by Zigmond and Snaith [27]. The German version of the questionnaire was used in this study [28]. It contains 14 items: seven questions measuring anxiety levels and seven items measuring depression levels. Each item is to be answered on a Likert scale that has four possible options (score ranging from 0 to 3). The maximum score that can be achieved in both the depression and anxiety scale is 21 with 0–7 denoting a normal test result, 8–10 points indicating mild depressive/anxious symptoms and more than 10 points representing severe symptoms [29].

### 2.4. Clinical Database

Baseline and perioperative data of all patients included in the study were retrospectively analyzed. Baseline characteristics comprised age, gender, body mass index (BMI in kg/m²), medical history, and American Society of Anesthesiologists (ASA) score. Perioperative data included the following outcome parameters: length of surgery, length of hospital stay, length of stay in the intensive care unit (ICU), postoperative complications including postoperative hemorrhage, bilioma/bile leak, surgical site infections (wound infections, fascia dehiscence, intraabdominal abscess), urinary tract infections, and pulmonary complications (pleural effusions, pneumothorax, and pneumonia). Complications were graded according to the classification published by Dindo and Clavien [30].

### 2.5. Statistics

All data were initially collected with Microsoft Excel (Microsoft, Redmond, DC, USA) and later analyzed using SPSS 25.0 (IBM, Armonk, NY, USA). Gaussian distribution was tested using the Kolmogorov-Smirnov test. Continuous variables were tested using nonparametric tests (Mann–Whitney U test and Kruskal–Wallis test) and are displayed as median (M) and range, except for the results of the questionnaire subcales, which are shown as mean and standard error of the mean. Categorical variables were compared using the Chi-square-test. A *p*-value < 0.05 was considered statistically significant.

## 3. Results

### 3.1. Laparoscopic Liver Resection for HCC: Patients’ Characteristics and Postoperative Outcomes

Ninety-eight patients underwent LLR for HCC between June 2014 and December 2019 and met the inclusion criteria for our survey. The median age was 70.5 (range 50–85) years, the majority of patients were male (*n* = 71, 72%). Most patients were multimorbid with at least two comorbidities (*n* = 78, 80%). Consequently, the general physical status of the majority of patients were classified as ASA 3 (*n* = 51, 53%). Liver cirrhosis as underlying liver disease was present in 72 patients (74%, Table 1).

Most patients were operated with conventional multiport laparoscopy (*n* = 64, 65%), followed by hand-assisted laparoscopic surgery (HALS, *n* = 25, 26%). Robotic surgery and single-incision laparoscopic surgery (SILS) was performed in 4 (4%) and 5 (5%) patients, respectively. Most hepatectomy procedures were considered minor resections (*n* = 83, 85%), defined as the resection of two segments or less. 

Overall morbidity was 25% (*n* = 25) with the majority of complications being minor complications (Dindo/Clavien I and II: *n* = 17, 17%). One patient died postoperatively from septic shock secondary to bilateral pneumonia (overall 90-day mortality 1%). Table 1 shows all patient characteristics, details of surgical techniques, extent of resection, and perioperative outcome parameters.

### 3.2. HRQoL (SF-36) and Mental Health (HADS) after Laparoscopic Liver Resection

The results of the SF-36 questionnaire from patients after LLR for HCC are displayed in Table 2. The median time between surgery and questionnaire was 15 (3–29) months. The findings were compared to those of the German general population [24]. The selected subset of 133 subjects was similar to the LLR group in terms of age (69 years; 38–79) and gender (female 36%, *n* = 48; male 64%, *n* = 85). 

Patients scored lowest in the dimensions Role-Physical, General Health, and Vitality. Compared to the general population, out of the physical component items, only General Health was rated significantly lower in HCC patients after LLR (LLR 52 (25–100) vs. general population 67 (15–97), *p* = 0.013), leading to a similar score in the Physical Component Summary (LLR 47 (21-59) vs. general population 47 (16–60), *p* = 0.442). In the mental component, on the other hand, patients with HCC after LLR had a significantly lower Component Summary score (LLR 51 (28–67) vs. general population 56 (23–72), *p* = 0.004). This difference was driven by lower scores in the items Social Functioning (LLR 87.5 (0–100) vs. general population 100 (0–100), *p* < 0.001) and Role-Emotional (LLR 100 (0–100) vs. general population 100 (0–100), *p* = 0.006), while there was only a trend towards lower Vitality, and Mental Health scores were similar to the general population.

The results of the HADS questionnaire in HCC patients after LLR are shown in Figure 1. With regards to anxiety, 21 (53%) of all patients had normal scores (<7), 14 (35%) had borderline abnormal scores (8–10), and five (13%) had abnormal scores. The overall mean score was 7 (0–17). On the depression subscale, 33 (83%) had normal scores (<7), one (3%) had a borderline abnormal score (8–10), and 6 (15%) had abnormal scores. The mean depression score was 4.5 (0–14). 

### 3.3. Factors Influencing HRQoL and Mental Health after Laparoscopic Liver Resection

To identify factors that might influence HRQoL and mental health after laparoscopic liver resection, various clinical factors were tested for their impact on the outcome of the survey. In a univariate analysis, the factors ASA score, presence of comorbidities, postoperative morbidity, age, gender, and liver cirrhosis did not influence the outcome of the surveys (all eight SF-36 subscales as well as summary scales for Physical and Mental Components, respectively; HADS subscales anxiety and depression, all *p* > 0.05, data not shown). Interestingly, the time span between LLR and completion of the questionnaires effected the SF-36 subscales General Health (*p* = 0.041), Vitality (*p* = 0.032), Role-Emotional (*p* = 0.011), and Mental Health (*p* = 0.025), with higher scores in the group that was questioned within 12 months after surgery when compared to later time points (Figure 2a). Similarly, there was a trend towards higher anxiety and depression scores among patients who had been operated more than 12 months ago (*p* = 0.050 and 0.107, respectively), with statistically significant differences in the HADS Total score (*p* = 0.038, Figure 1a). 

### 3.4. Laparoscopic Liver Resection vs. Liver Transplantation for HCC—Impact on HRQoL and Mental Health

The historic cohort of 70 patients who underwent OLT for HCC had a median age of 63 (range 49–77) years; the majority of patients was male (*n* = 59, 83%), with no significant difference in age or sex to the LLR cohort. Median time since OLT was 27.5 (2–185) months. With regards to comorbidities, 35 (50%) had (compensated) kidney failure, 45 (64%) had diabetes mellitus, and 59 (84%) had arterial hypertension. 

Compared to HCC patients after OLT, those who underwent LLR had significantly lower scores on the General Health scale (*p* = 0.029). Social Functioning was lower in the LLR group as well; however, the difference was short of statistical significance (*p* = 0.066). The other subscales for HRQoL were similar in both groups. The results of the HADS questionnaire revealed significantly higher anxiety scores for the LLR group (*p* = 0.010), which also led to a significantly increased HADS summary score (*p* = 0.041) when compared to patients after OLT. The comparison of HRQoL and mental health of patients after LLR and OLT is displayed in Figure 1b (HADS) and 2b (SF-36).

## 4. Discussion

In the present study, we show that LLR for HCC is safe with low complication rates, despite high rates of liver cirrhosis in HCC patients. Furthermore, patients who undergo LLR for HCC report a good quality of life, comparable in most aspects to the general population and to HCC patients after OLT. 

Liver resection is increasingly performed in minimal invasive approaches and has been found to be safe and render comparable oncologic results to open surgery, with some favorable short-term perioperative outcome parameters [12,31,32]. We have previously published results from a small observational HRQoL study that included patients after liver resections for any indications and saw no significant differences between LLR and open liver resection (OLR) [22]. Interesting results were also published recently from a randomized controlled trial for LLR vs OLR for colorectal liver metastasis, which showed that HRQoL was significantly better after LLR when compared to OLR, and that the difference is still measurable four months after surgery [22,33]. However, none of these studies have focused on the unique population of HCC patients. As laparoscopic liver resection has been proposed to be favorable in patients with cirrhosis [34,35,36], the research on outcomes for HCC patients especially after LLR has to be extended to the study of HRQoL. 

An older study compared patients who had HCC in liver cirrhosis with non-HCC cirrhotic patients and found that HRQoL is reduced in cirrhotic patients when compared to the general population in all dimensions of SF-36, and was only significantly more impaired in HCC patients in Role-Physical and Bodily Pain [37]. Other studies compared HRQoL of HCC patients before and after treatment with TACE showing that HRQoL improved after TACE, but only after two and four months, while later time points fail to display this improvement [38,39]. Interestingly, compared with their population of HCC patients, our patients have substantially higher scores in all dimensions of the SF-36. This difference between levels of HRQoL between the studies might correlate with a certain selection bias in their cohorts towards a more advanced disease stage [38,39], as their patients were obviously not considered for surgical treatment.

While the above-mentioned studies have focused on HRQoL of HCC patients at baseline or after non-surgical treatments, there are a few studies that have also evaluated HRQoL of HCC patients before and after liver resection and have found that HRQoL is generally good after surgical treatment [40,41]. A more recent study from Taiwan tested HRQoL in HCC patients before and at three and six months after surgical resection and found that patients had significantly improved HRQoL measures at three months postoperatively, which stayed the same at six months [17]. However, this latter study does not disclose the techniques of surgery, and the other quoted studies only included patients after OLR. Our results are therefore an important piece of evidence linking postoperative HRQoL of HCC patients to minimal invasive liver surgery.

Our study is a cross-sectional study with one time point of data collection. This poses a limitation because we do not have pre-operative HRQoL data from our patients. However, many other longitudinal studies have shown that HRQoL is reduced initially after liver resection, but improved when compared to pre-operative status after 1–3 months [17,38,39]. These changes seem to generally be long-lasting, except in the case of recurrence. Our patients were questioned at different time points after surgery. Interestingly, we observed a correlation of HRQoL outcomes and the time that had passed after surgery: patients who were questioned after 12 months or more had significantly lower scores in the items General Health, Vitality, Role-Emotional, and Mental Health. These patients also scored higher on the HADS score when compared with the more recently operated. This may be explained by the persisting underlying disease of liver cirrhosis which, untreated, is likely to progress over time. Apart from this correlation with time after surgery, no other clinical parameter, including the presence of liver cirrhosis, was found to be associated with postoperative HRQoL. Interestingly, no difference in HRQoL was seen between male and female patients in our cohort, although female sex has been implicated with lower HRQoL both in the general population and in patients with chronic liver disease [42,43]. However, a gender difference may not be detectable in our cohort because of the small number of female patients. 

We compared our HRQoL findings from patients after LLR for HCC with the general population, showing that except for three subscales (General Health, Social Functioning and Role-Emotional), the results are similar. Of course, this control group is not ideal, as all our patients have HCC and most have underlying liver cirrhosis, which are rare in the general population and are likely to impair HRQoL by themselves. However, this comparison suggests that negative effects of LRR on HRQoL are minor. It is plausible that the mental health subscale of Social Functioning and Role-Emotional might be impacted more by the underlying disease than the surgery. 

In comparison with the other curative treatment option of OLT, we here show that HCC patients after LLR display slightly lower General Health perception than patients after OLT, while otherwise showing similar levels of HRQoL. Furthermore, patients after LLR seem to have higher anxiety levels. This finding may again be explained by the fact that resection of the tumor does not cure the patients of their underlying liver cirrhosis and patients are aware of the high risk of tumor recurrence. A study by Lei et al. compared HRQoL after liver resection and OLT for HCC [44], but the authors do not disclose the resection technique. According to the time frame (2000–2010) it is rather more likely to include predominantly OLR. The authors find no difference of HRQoL between patients after resection and OLT, but importantly, they completed their survey with considerably more delay to surgery (at least three years) than in our study. Therefore, it is possible that initial differences that we observe might be levelled after some time. However, a complete longitudinal study would be necessary to answer that question, since in the study of Lei et al, using such a long time point after surgery (mostly more than five years) might cause a selection bias by eliminating patients who die before that [44].

The comparison of our LLR patients with the historic cohort of patients after OLT has to be interpreted with caution, as these two cohorts were treated and surveyed during different time points and are unmatched due to relatively small group sizes. However, while the assessment of HRQoL of HCC patients after LLR is the main focus of this study, we do believe that comparison with OLT is interesting for the evaluation of surgical treatment strategies for HCC, especially since the two treatments differ immensely in invasiveness.

## 5. Conclusions

In summary, patients with HCC report a good quality of life after laparoscopic liver resection. Values for the dimensions General Health, Vitality, Role-Emotional, and Mental Health were higher when questioned in the first 12 months following surgery when compared to later time points. Compared to patients who received OLT for HCC, General Health perception was significantly lower in patients after LLR, combined with higher anxiety levels, possibly due to the persistence of underlying liver cirrhosis. Taken together, we here show that LLR is not only a safe treatment option for patients with HCC, but also results in good postoperative quality of life. Especially in times of organ shortage and without the option of transplantation, LLR should always be considered an option for resectable HCC. 

## Figures and Tables

**Figure 1 healthcare-09-00694-f001:**
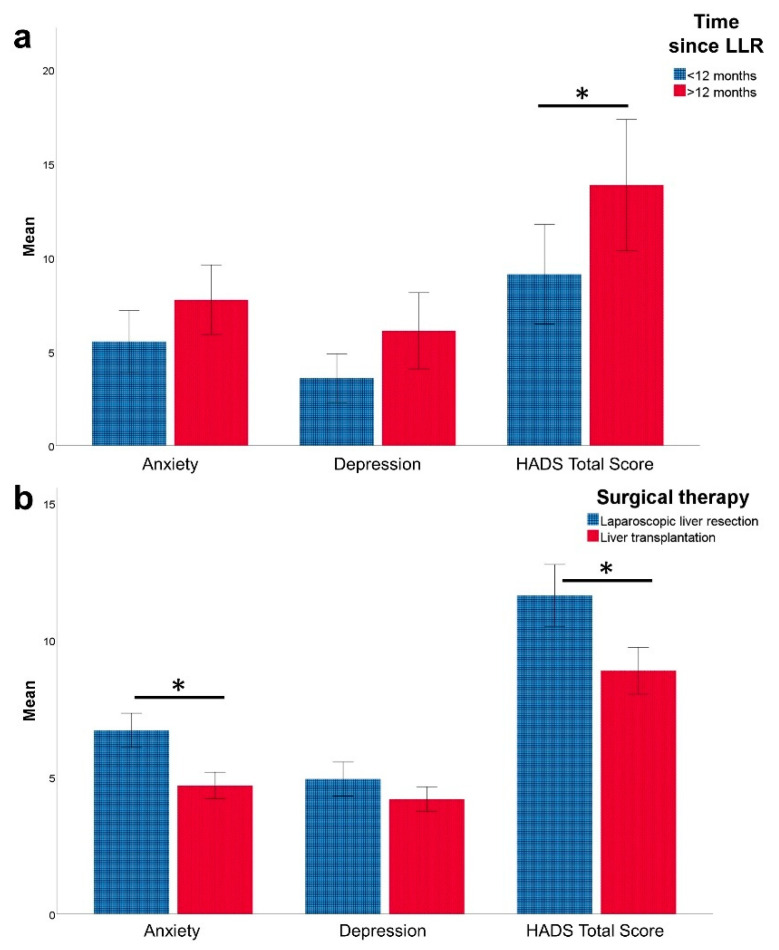
Mental health measured with Hospital Anxiety and Depression Scale (HADS) in patients with hepatocellular carcinoma (HCC) after undergoing laparoscopic liver resection (LLR). Error bars = standard error of the mean. * *p* < 0.05. (**a**) Mental health (HADS) in patients after LLR according to the time between LLR and completion of the questionnaire. (**b**) Comparison of mental health (HADS) of HCC patients after LLR and after orthotopic liver transplantation (OLT).

**Figure 2 healthcare-09-00694-f002:**
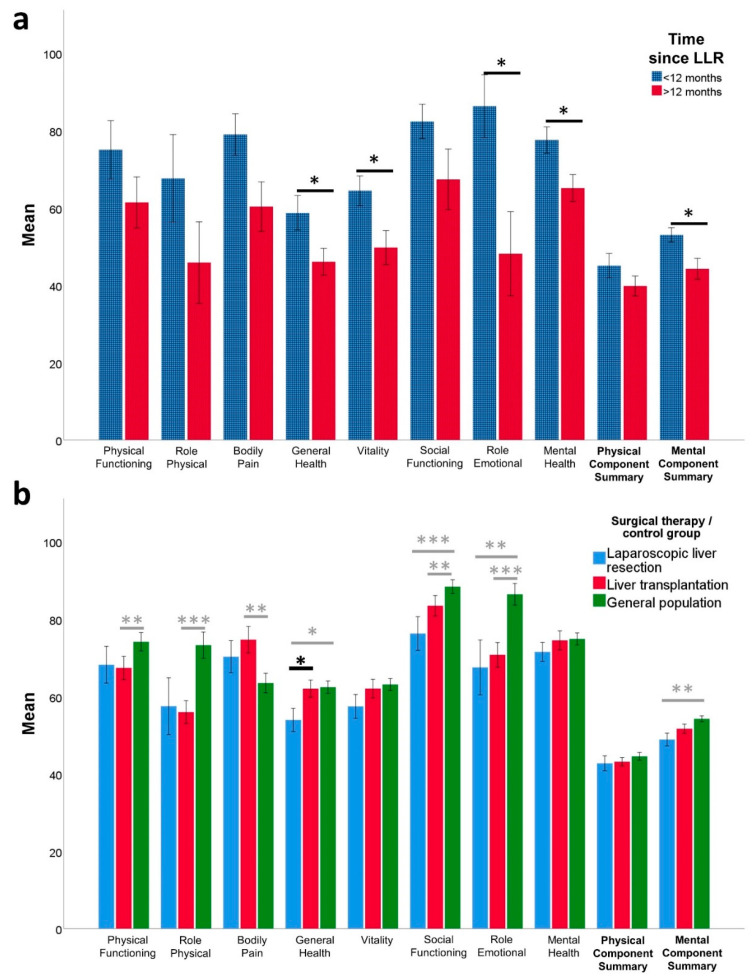
Health-related quality of life measured with Short Form 36 (SF-36) in patients with hepatocellular carcinoma (HCC) after undergoing laparoscopic liver resection (LLR). Error bars = standard error of the mean. * *p* < 0.05, ** *p* < 0.01, *** *p* < 0.001; (**a**) Health-related quality of life (SF-36) according to the time between LLR and completion of the questionnaire; (**b**) Comparison of health-related quality of life (SF-36) of HCC patients after LLR and orthotopic liver transplantation and the German general population (matched for age and gender). Differences between HCC patients after either surgical therapy and the general population are labeled in grey; differences between LLR and OLR are labeled in black.

**Table 1 healthcare-09-00694-t001:** Patients’ characteristics, surgical techniques, and perioperative outcomes after laparoscopic liver resection for hepatocellular carcinoma (HCC).

	Laparoscopic Resection of Hepatocellular Carcinoma
	*n* = 98
Age ^a^	70.5 (50–85)
Body mass index (kg/m^2^) ^a^	26.5 (18–45)
Gender (male) ^b^	71 (72)
Presence of liver cirrhosis ^b^	72 (74)
Missing	1 (1)
ASA score for physical status ^b^	
<3	34 (35)
≥3	52 (53)
Missing	12 (12)
Surgical technique ^b^	
Multiport laparoscopy	64 (65)
Hand assisted laparoscopy	25 (26)
Single incision laparoscopy	5 (5)
Robotic surgery	4 (4)
Extent of resection ^b^	
Minor liver resection	83 (85)
Major liver resection	15 (15)
Length of operation (minutes) ^a^	209 (49–461)
Complications (Clavien-Dindo) ^b, c^	
None	73 (75)
I–II	17 (17)
III	7 (7)
IV–V	1 (1)
Intensive Care Unit stay (days) ^a^	1 (0–8)
Hospital stay (days) ^a^	9 (3–27)

^a^ Data is presented as median (range).^b^ Data is presented as number (percent) [30].

**Table 2 healthcare-09-00694-t002:** Health-related quality of life measured with Short Form 36 questionnaire (SF-36) after laparoscopic liver resection for hepatocellular carcinoma (HCC).

	Laparoscopic Resection of Hepatocellular Carcinoma
	*n* = 43 *
Physical Functioning	80 (5–100)
Role Physical	85 (0–100)
Bodily Pain	64 (21–100)
General Health Perceptions	52 (25–100)
**Physical Component Summary**	**47 (21–59)**
Vitality	55 (10–90)
Social Functioning	87.5 (0–100)
Role Emotional	100 (0–100)
Mental Health	70 (40–100)
**Mental Component Summary**	**51 (28–67)**

Data is presented as median (range). * All patients who returned the questionnaire.

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
