# Peer review of "Health-Related Quality of Life and Mental Health after Surgical Treatment of Hepatocellular Carcinoma in the Era of Minimal-Invasive Surgery: Resection versus Transplantation"

_healthcare, 2021, doi:10.3390/healthcare9060694_

Round 1

Reviewer 1 Report

In this study, the authors compared the health-related quality of life (HRQoL) and mental health between orthotopic liver transplantation (LLR) and orthotopic liver transplantation (OLT) in hepatocellular carcinoma patients to determine the feasibility of LLR for the replacement of OLT. This suggests the possibility that LLR may be in the spotlight as an alternative to transplant surgery (OLT) due to a shortage of organs. However, there are major issues that need to be addressed in this study.
1. line 36-37: cirrhosis is a pre-stage of liver cancer. However, the author is aware of it as a complication due to cancer progression. A clear theoretical background from normal liver to HCC progression including middle-stage, such as cirrhosis, should be provided.
2. It is generally known that LLR is usually a priority for liver disease, of course excluding unresectable conditions. However, the author argues that OLT is being considered first for liver disease, especially in HCC, instead of LLR. What is the rationale or basis for that? This could be an important issue where the contents in the introduction may need to be revised overall.
3. Line 60: Is this true? There are so many reports for QoL in patients undergoing liver resection.
4. Study design: 
- The authors revealed that all patients underwent LLR, if so, it is wondering how QoL and mental health between LLR and OLT were compared.
- Specific items or methods for the survey and clavien-dindo classification (table 1) must be provided as supplementary.
5. What does Figure 1A mean? Is only the score of the LLR group represented by a bar graph? Such a presentation is very awkward because of without a comparison group. It is wondering how the results of the OLT group in Figures 1b and c were gathered.
6. There is the same issue with data collection targeting LLR and general populations in Figures 2a and b.
7. In Figure 2b, general health is significantly improved following OLT. Surprisingly, the mean value was almost the same with general populations. Does the author believe that LLR is a clear operation replacement for OLT?
8. Overall contents: As pointed out in comment #2, since OLTs are considered in situations where LLR is impossible, it will have to be revised the contextual statements that are inferred that LLRs can replace OLT in any conditions.

Reviewer 2 Report

In this manuscript, the authors made a comparison for patients with resectable hepatocellular carcinoma (HCC) receiving laparoscopic liver resection (LLR) or orthotopic liver transplantation (OLT), in terms of their post-operative Health-related quality of life (HRQoL) and mental health. The study is meaningful since it will provide information for doctors to decide the treatment for their HCC patients, especially when considering the shortage of organ donation and improved minimally invasive surgery technology. For the above reasons, I support the publication of this work, and the following points should be addressed.

  1. In your study design, in line 91, you are comparing the collected result with HRQoL from a subset of the German general population. The comparison itself is ok but the data of the German general population you use were taken between 1997 and 1999, which is outdated and should be replaced with some updated data for a more meaningful comparison.
  2. For the patient data who underwent OLT for HCC, could you generate a similar table to Table 1, or add extra column in Table 1 so the readers will have a better idea of the compared group?
  3. Although the male patients is the majority of the studied group, can you comment on the gender difference between the two groups in your discussion part?

Round 2

Reviewer 1 Report

Thank you for considering my comments. That was a great response and improvement that dispelled this reviewer's doubts.